# Ambient PM Concentrations as a Precursor of Emergency Visits for Respiratory Complaints: Roles of Deep Learning and Multi-Point Real-Time Monitoring

SungChul Seo [1] , Choongki Min [2], Madeline Preston [3], Sanghoon Han [2], Sung-Hyuk Choi [4] , So Young Kang [5] and Dohyeong Kim [3],*

1  Department of Nano, Chemical and Biological Engineering, Seokyeong University, 124 Seokyeong-ro, Seongbuk-gu, Seoul 02713, Korea; haha0694@gmail.com
2  Waycen, 9 Yeongdong-daero 75-gil, Gangnam-gu, Seoul 06182, Korea; choongki.min@waycen.com (C.M.); sanghoon.han@waycen.com (S.H.)
3  School of Economic, Political and Policy Sciences, University of Texas at Dallas, 800 W Campbell Rd., Richardson, TX 75080, USA; map170000@utdallas.edu
4  Department of Emergency Medicine, Korea University Guro Hospital, 148 Gurodong-ro, Guro-gu, Seoul 08308, Korea; kuedchoi@korea.ac.kr
5  Department of Police Science, Konkuk University, 268 Chungwon-daero, Chungju 27478, Korea; kkangssyy@kku.ac.kr
*  Correspondence: dohyeong.kim@utdallas.edu

**Abstract:** Despite ample evidence that high levels of particulate matter (PM) are associated with increased emergency visits related to respiratory diseases, little has been understood about how prediction processes could be improved by incorporating real-time data from multipoint monitoring stations. While previous studies use traditional statistical models, this study explored the feasibility of deep learning algorithms to improve the accuracy of predicting daily emergency hospital visits by tracking their spatiotemporal association with PM concentrations. We compared the predictive accuracy of the models based on PM datasets collected between 1 December 2019 and 31 December 2021 from a single but more accurate air monitoring station in each district (Air Korea) and multiple but less accurate monitoring sites (Korea Testing & Research Institute; KTR) within Guro District in Seoul, South Korea. We used MLP (multilayer perceptron) to integrate PM data from multiple locations and then LSTM (long short-term memory) models to incorporate the intrinsic temporal PM trends into the learning process. The results reveal evidence that predictive accuracy is improved from 1.67 to 0.79 in RMSE when spatial variations of air pollutants from multi-point stations are incorporated in the algorithm as a 9-day time window. The findings suggest guidelines on how environmental and health policymakers can arrange limited resources for emergency care and design ambient air monitoring and prevention strategies.

**Keywords:** particulate matters; emergency visits; deep learning; multi-point monitoring

In the last 30 years, the global prevalence of chronic respiratory diseases has increased by nearly 40% and remains a leading cause of death around the world [1,2]. One of the many factors contributing to respiratory diseases is ambient air pollution [3,4]. According to the 2017 Global Burden of Disease (GBD) Study, there were 3.2 million deaths due to chronic obstructive pulmonary disease (COPD) and 495,000 deaths due to asthma [5]. People visit emergency departments for various reasons, but ambient air pollutants are a major risk factor as they can impact the respiratory and cardiovascular systems. Several studies have identified an association between hospital admissions and emergency room visits for respiratory and cardiovascular diseases due to various ambient air conditions [4,6–9]. Most of these studies utilized linear statistical models or time series forecasting methods to obtain evidence that hospital admissions and emergency room visits for respiratory and

cardiovascular diseases were significantly correlated with particulate matters ($PM_{10}$ and $PM_{2.5}$) [4,6–8] and other pollutants such as ozone ($O_3$), nitrogen dioxide ($NO_2$), carbon monoxide (CO) and sulfur dioxide ($SO_2$) [4,9].

Given the advancement of machine learning algorithms and processes, recent works have contributed to this literature by examining how deep learning models can be used to improve predictions for diseases triggered by ambient air pollution [10–13]. These learning techniques are important for the accurate prediction of hospital admissions or emergency room visits as they could help optimize the allocation of scarce medical resources. Artificial neural networks (ANN) are an attractive approach for understanding air pollution's impact on human health since they are better suited to deal with nonlinear mapping problems compared to linear approaches, while traditional methods may lose prediction accuracy in situations where multiple factors fluctuate in a nonlinear fashion [14]. Moreover, deep learning algorithms can process high-dimensional big data samples better than traditional statistical modeling, which could be more suitable for small datasets. Recently, Lu et al. [10] used three different machine learning models to predict emergency room visits in Beijing, China, showing that the LSTM model reaches the highest accuracy with an advantage in the detection of a lag effect. Tadano et al. [11] used two classic ANN architectures, extreme learning machines (ELM) and echo state networks (ESN), to expand the literature on ML applications for predicting respiratory-related hospital admissions due to $PM_{10}$ concentrations, indicating that the ELM model was the best predictor of daily hospital admissions for respiratory diseases given the $PM_{10}$ concentration, relative humidity, and ambient temperature inputs.

Despite growing concern about air pollution in South Korea, particularly particulate matter, only a few studies have examined the relationship between ambient air pollution and respiratory diseases [15–17]. Son et al. [17] and Han et al. [15] found that exposure to $PM_{10}$, $PM_{2.5}$, CO, $O_3$, $NO_2$, and $SO_2$ may lead to increased hospital admissions for various respiratory conditions. Nakao et al. [16] concluded that PM, $NO_2$, and $O_3$ caused respiratory symptoms that led to the deterioration of health-related quality of life. However, most of the existing studies in South Korea utilized traditional statistical models to show which environmental pollutants are associated with hospital admissions or emergency room visits for respiratory diseases, with less attention paid to spatial variations of air pollutants within a monitoring region. They tend to take air pollution measurements from one or only a few monitoring stations to calculate the daily averages of various pollutants, assuming all residents within the region are exposed to the same level of ambient air pollutants. However, measurements vary between stations and could lead to errors or biases when averaging, and thus the process of predicting emergency room visits due to respiratory illnesses could be improved by incorporating multipoint air pollution data into an advanced learning algorithm [11].

Therefore, this paper aims to contribute to the literature by developing a predictive algorithm based on real-time PM data from multiple monitoring locations in a district of Seoul, with special attention paid to the following questions:

- Would a deep learning algorithm be a feasible tool to improve the accuracy of predicting emergency hospital visits by tracking spatial and temporal PM trends?
- Would predictive accuracy in a deep learning model be substantially improved when air monitoring data come from multi-point locations?

For these aims, we compared the routine ambient air quality monitoring data provided by the Korean Ministry of Environment titled "Air Korea" with new sensor-based air monitoring data from the Korea Testing & Research Institute (KTR). For the Air Korea data, $PM_{2.5}$ concentrations were measured by the standard β-ray measurement method at 5 min intervals and then aggregated into hourly averages through statistical processing before being stored in a database [18]. This government-grade air monitoring station provides relatively accurate data but is expensive to run due to high installation and maintenance costs [19]. A total of 522 "Air Korea" stations are installed across South Korea as national air quality monitoring stations (AQMS) [20], with generally one station in each urban

district [10]. In Seoul, one Air Korea station covers about 15.5 km$^2$ on average, which generates wide holes between different monitoring stations and hampers the localized assessment of dynamic trends of air pollutants within the region [21]. We thus developed a new sensor-based air monitoring instrument with a much cheaper cost and installed a total of 24 instruments with approximately 1 km$^2$ scale resolution in the same district [21]. It is designed to measure PM$_{2.5}$ concentration through the light scattering method while maintaining a temperature of 20–30 °C and relative humidity of less than 70% with a pretreatment control device. To validate the accuracy of the module, a co-location test was performed with a reference monitor (i.e., a beta attenuation monitor), showing a coefficient of determination of 0.957 [21].

The real-time ambient PM$_{2.5}$ (μg/m$^3$) and PM$_{10}$ (μg/m$^3$) measurements, as well as temperature (°C) and humidity (%), were obtained from the two databases, Air Korea and KTR, between 1 December 2019 and 31 December 2021 (697 days) only in Guro District in Seoul, South Korea. The air monitoring data were collected every 5 min but aggregated to daily average values. As seen in Figure 1, the Air Korea station is located near the eastern boundary of the district, while the 24 KTR stations are scattered throughout the district. The daily emergency visit data due to respiratory symptoms (N = 697) were obtained for the same period from the Korea University Guro Hospital, the only regional emergency center located within the district, and integrated with the PM dataset (N = 200,736 for every station). The dataset includes only those patients who were diagnosed with a respiratory disease based on the ICD-10 code. Patient address information could not be accessed due to confidentiality concerns, but the hospital confirmed that about half of their patients lived in Guro District and nearly all patients came from the vicinity of the hospital for emergency visits. The locations of all the KTR monitoring stations and the hospital are geocoded to calculate a Euclidean distance (km) to the hospital from each station, which is used as a weight factor when the PM data from multiple locations are integrated.

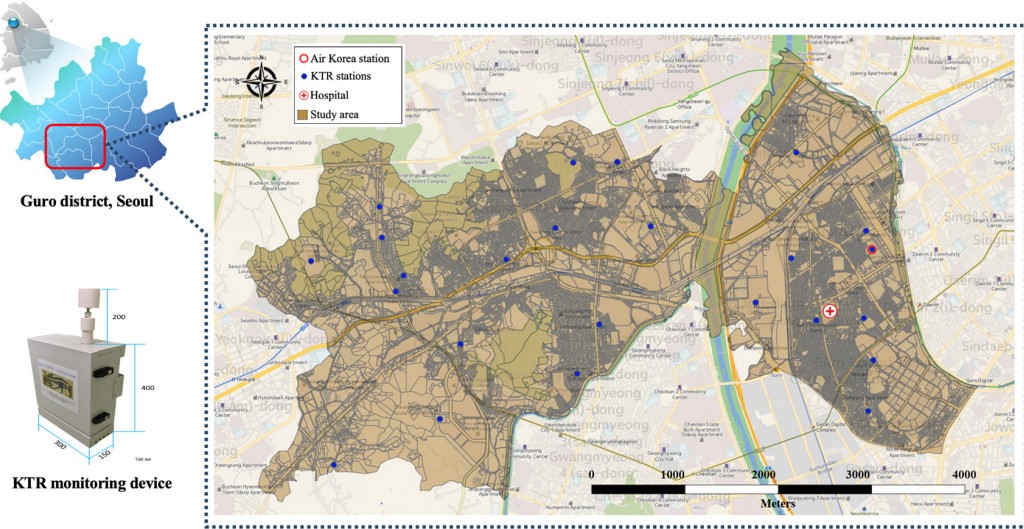

**Figure 1.** Locations of Air Korea and KTR monitoring stations and emergency hospital within the study area.

The database included the daily PM measurements, temperature and humidity from Air Korea and KTR stations, as well as daily emergency department visit counts due to respiratory complaints and the distance among them. Once constructed, it was used to develop a predictive model that can identify the expected number of emergency visits for the next day based on the previous N days (timesteps) of monitoring data. A two-step predictive model is introduced here via the MLP (multilayer perceptron) and LSTM (long short-term memory) network algorithms applied sequentially. Initially, the MLP was designed to estimate distance-weighted PM measurements from multiple KTR stations as

one integrated feature structure at different timesteps, which were used as input data in the LSTM network. The MLP algorithm contains five layers, including one input layer for each of the 24 stations, one output layer representing a weighted score at various time steps, and three hidden layers. The weighted score indicates the level of relative contribution to the output variable of each input factor from each station located at various distances to the hospital. The output from MLP was transferred to the LSTM algorithm, which has two learning layers to predict daily emergency visit counts within the district, contingent on different time steps of 1, 3, 6 and 9 days. The LSTM algorithm was used to capture complex temporal trends, including seasonality, because it should be able to model non-linear relationships in the data, while a traditional time-series model typically assumes the future value of a variable is assumed to be a linear function of several past observations and random errors. The model with a larger time step assumes an extended cumulative effect of air pollution on adverse outcomes of respiratory symptoms, while the model using one-day time steps focuses only on the impact of PM exposure during the previous 24 h. For both algorithms, the first 90% of the records (from Day 1 to 627) were used for training to 7000 epochs, while the last 10% of the data (from Day 628 to 697) were used to test and compare the relative performance of the algorithm based on RMSE from each model (Air Korea vs. KTR) and each time step (1, 3, 6 and 9 days). Deep learning and statistical analyses were performed by using Python 3.9.7, TensorFlow 2.6.0 and scikit-learn 1.0.1, and visualization was performed using Python's matplotlib library 3.4.3. The data and source codes are available upon request. The entire structure of the 2-step deep learning procedure is depicted in Figure 2.

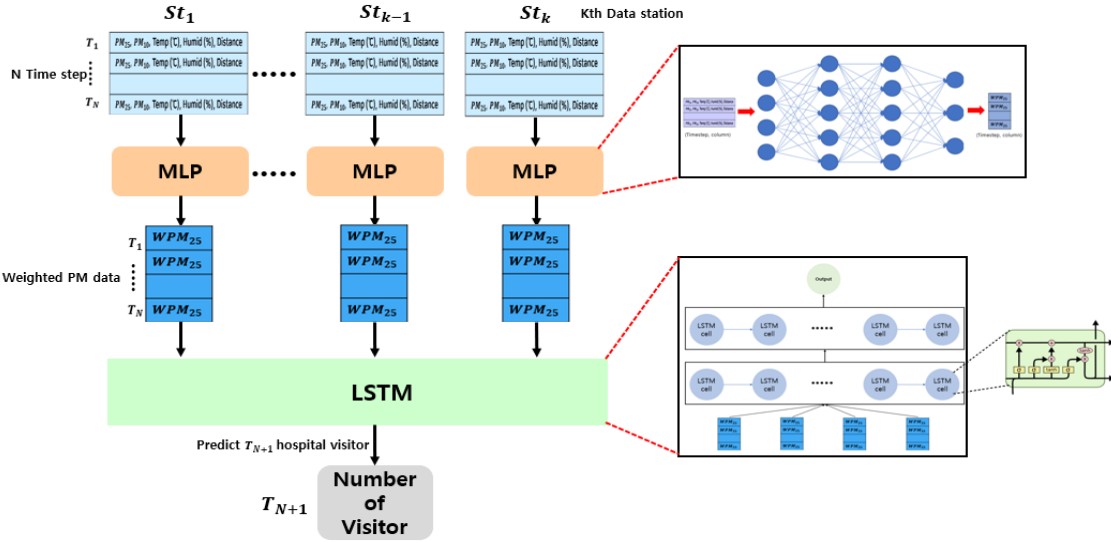

**Figure 2.** Two-step deep learning structure processing data from multi-point monitoring stations.

Figure 3 illustrates substantial daily fluctuations in observed counts of emergency room visits during the training period—the first 627 days (blue line). For the last 70 days, the observed counts (orange line) are compared with the predicted counts based on two different datasets (Air Korea vs. KTR) and two different time steps (1-day window in green vs. 9-day window in red). It appears that the predicted model failed to perform well on either dataset when only one days' worth of data in the past were used for training and prediction. However, the predictive accuracy seems substantially improved when the model was trained based on the historical trends of PM concentrations during the past 9 days. These results indicate that the Air Korea model is not well suited for an accurate prediction of emergency visit counts due to respiratory symptoms regardless of the time span used, while the prediction performance of the multipoint KTR model improves when the temporal window is expanded to include more data from the previous days.

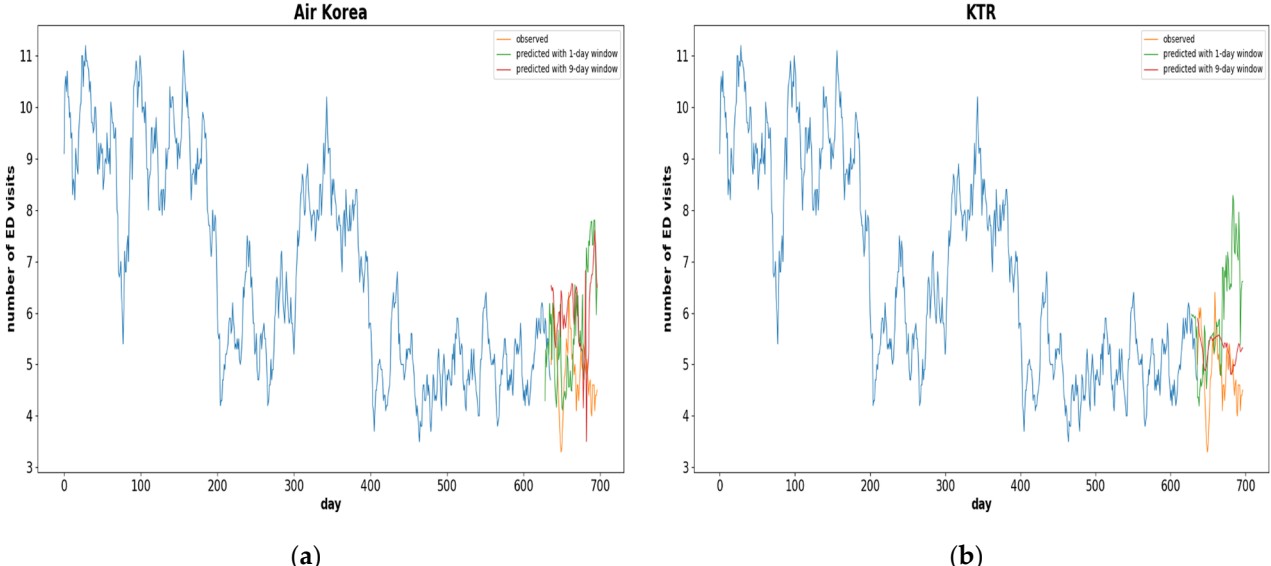

**Figure 3.** Observed and predicted counts of emergency department visits by different time steps. (**a**) Prediction by PM data from Air Korea; (**b**) Prediction by PM data from KTR.

Figure 4 demonstrates that the single-stage model based on the Air Korea data experiences only a minimal change and even an increase in RMSE as the time window increases, with the highest error occurring in the 9-day model. In comparison, the multipoint station model based on KTR shows a continuous reduction in RMSE by increasing the learning timespan for the training network from 1 day to 9 days (1.67 to 0.79). Figure 5 shows scatterplots showing the relationship between observed and predicted counts of emergency visits based on each model (Air Korea vs. KTR) using the 9-day window, along with R-squared values (0.284 vs. 0.864). The improved predictive accuracy of the KTR model is likely to be attributed to multi-point PM data collection throughout the study area, clearly highlighting the importance of incorporating the spatial variation in air pollutants in the predictive modeling algorithm. This finding also underscores the role of a longer period of air monitoring and model training that would capture the impact of cumulative exposure to air pollutants in predicting the demand for related health services.

This study assessed the feasibility of a deep learning-based predictive algorithm to gauge the demand of daily admission counts to an emergency department due to respiratory complaints associated with PM exposure using two different real-time databases (Air Korea as a single point but with more accurate monitoring vs. KTR as multi-points but with wider coverage) for a district of Seoul, Korea. By employing two-step deep learning techniques, MLP and LSTM, rather than traditional statistical models, we were able to capture the complex interconnectivity between various input layers from multiple stations and the daily fluctuation of output values. We also found that integrating the PM database with multipoint monitoring stations resulted in a decrease in RMSE when the time window for learning was increased, reflecting that tracking cumulative PM exposure for a longer time period could improve prediction accuracy only if the data contain sufficient spatial variation of input features. The study was able to determine that there was no improvement in the prediction processes with the increase in the time window when the data were collected from a single-point monitoring station. The results are in line with the literature that deep learning algorithms tend to perform better when more spatial and temporal data are incorporated into the learning process [22]. The findings would provide the guidelines for environmental and health policymakers to proactively arrange or manage available medical resources and ambulances more appropriately based on the accurate prediction of daily emergency room demand. The lessons from this study also have implications for the effective design of PM monitoring strategies such as a long-

term tracking of multiple air pollutants from densely located monitoring stations with a low-cost sensor.

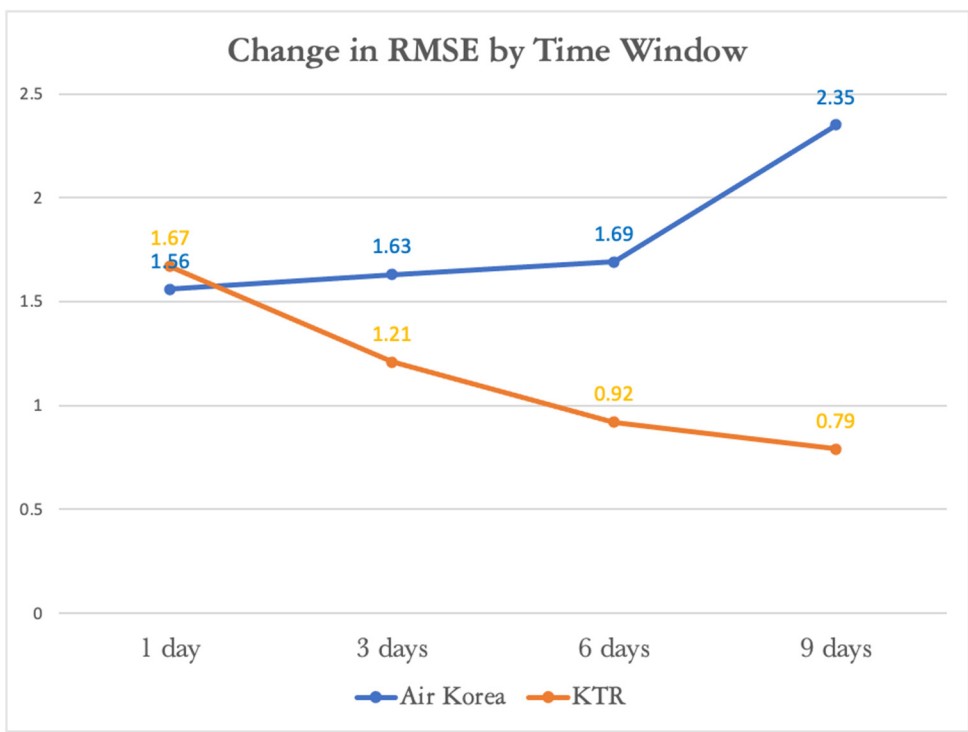

**Figure 4.** Comparison of predictive accuracy by time window between Air Korea and KTR Data.

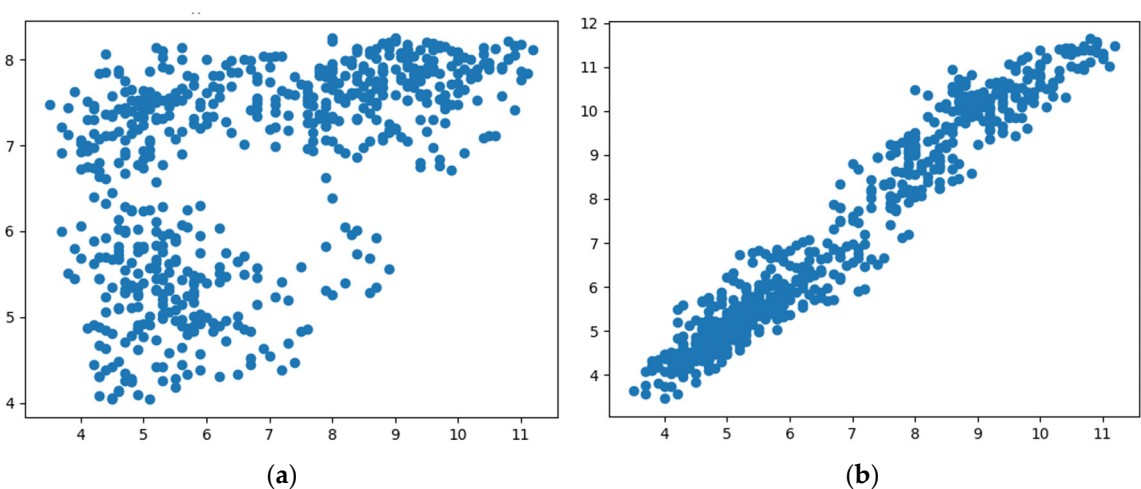

**Figure 5.** Scatterplot on the relationship between observed and predicted counts of emergency department visits using 9-day window: (**a**) Air Korea (R-squared: 0.284), (**b**) KTR (R-squared: 0.864).

It is worth noting that some limitations are inherent to this approach. First, this study ignored indoor PM exposure but simply assumed they have similar spatiotemporal patterns with outdoor exposure. Given that indoor PM exposure could have much larger variability and be more harmful than outdoor air quality [23], indoor PM concentration must be also monitored and included in learning and predicting processes, along with other indoor environmental characteristics such as access to air conditioning or ventilation systems, secondhand smoke exposure and so forth. Second, this study used only temperature and humidity as environmental covariates of PM exposure for predictive modeling, whereas other environmental pollutants can also contribute to or exacerbate respiratory symptoms. Third, due to the lack of patient-level data, this study failed to incorporate not only their

actual exposure level to PM based on their residential addresses and mobility patterns, but also their sociodemographic, physical and behavioral characteristics. Lastly, our predictive models (both Air Korea and KTR) may not perform well in predicting the acute effect of PM on ER visits, partly due to the small sample size. For the 1-day window model which can potentially detect acute effects of PM, only 576 data points (=24 h * 24 stations) were used for training, which may not be sufficient. In addition, we found that PM concentration levels were not so high during the prediction period and so the acute effect could be insignificant during that time. Future research should examine how the inclusion of more relevant variables collected from a larger study area can improve prediction accuracy, particularly those measuring individual levels of exposure.

**Author Contributions:** Conceptualization, D.K. and S.S.; methodology, D.K. and C.M.; validation, D.K., C.M. and S.S.; formal analysis, C.M. and S.H.; resources, D.K., S.-H.C. and S.S.; data curation, C.M., S.-H.C. and S.H.; writing—original draft preparation, D.K., M.P., C.M. and S.S.; writing—review and editing, D.K. and M.P.; visualization, C.M. and S.H.; supervision, D.K. and S.S.; project administration, D.K., S.Y.K. and S.S.; funding acquisition, S.S. All authors have read and agreed to the published version of the manuscript.

**Funding:** This work was supported in part by the Environmental Health Action Program (Development of receptor-based environment-induced diseases prevention and management system using real-time collected environment and health information) under Project 2018001350005.

**Institutional Review Board Statement:** Not applicable.

**Informed Consent Statement:** Not applicable.

**Data Availability Statement:** Not applicable.

**Conflicts of Interest:** The authors declare no conflict of interest. The funders had no role in the design of the study; in the collection, analyses, or interpretation of data; in the writing of the manuscript, or in the decision to publish the results.

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
