# Peer review of "Ambient PM Concentrations as a Precursor of Emergency Visits for Respiratory Complaints: Roles of Deep Learning and Multi-Point Real-Time Monitoring"

_sustainability, doi:10.3390/su14052703_

Round 1

Reviewer 1 Report

The authors explored the feasibility of deep learning algorithms to improve the accuracy of predicting ER visits taken into account spatiotemporal distribution of PM. This study is interesting, but a few concerns need to be addressed.

Author Response

Thank you for your careful review and consideration of our manuscript. The comments were very helpful in further refining and substantially improving the manuscript. Please find our response letter along with the revised manuscript.

Reviewer 2 Report

Thank you very much for inviting me to review this manuscript, I am happy to review it. I have several suggestions for authors to be considered.

  1. In abstract: you need to mention what is the dataset you used in this study, daily or monthly data?
  2. The Abstract should be rewritten considering the time period and quantitative results of the study.
  3. How do you obtain Emergency Visits for Respiratory Complaints data? The dataset provided by the organization was based on any criteria of disease diagnosis? What is the residence respiratory disease patient? Are they live in Guro District in Seoul, South Korea? How can you know that? You need to justify your data; otherwise, your model is predicting biased results.
  4. Please add Material and Method section.
  5. I am unable to find the study area description of the study. This should be mentioned in the Material and Method section.
  6. You need to mention the statistical software you used to analyze all of your data in Material and Method section.
  7. Please add Results and Discussion heading
  8. How the author has incorporated distance along with other parameters used in this study. The distance selection criteria should be clearly mentioned.
  9. Line 133: The MLP algorithm contains five layers that accounts for one input layer for each of the 24 stations, one output layer representing weighted PM measurement at each station at various time steps, and three hidden layers. What criteria are used by the algorithm to incorporate multiple stations input and to develop only one weighted PM measurement?
  10. Give full spellings of COPD in the introduction and any other abbreviations that appear for the first time in the paper.
  11. The results depict an overall prediction of patients in Guro District in Seoul, South Korea. I suggest the authors to present the prediction results in map format to examine the spatial variations of emergency department visits. This would be more meaningful to asses the hotspot areas of emergency department visits.
  12. Line 105: To validate the accuracy of the module, a co-location test was performed with reference monitor (i.e. beta attenuation monitor), showing a coefficient of determination of 0.957. I suggest authors to display this result along with a scatterplot.

Author Response

(The authors gave the same response as above.)

Reviewer 3 Report

GENERAL OVERVIEW: the authors presented a manuscript (Communication) regarding the use of deep learning approach and multipoint-RT monitoring to understand the relationship between PM and related respiratory diseases. Nowadays, this is a very interesting topic, mainly due to other associated respiratory disease risks in an indoor environment. The results are very promising, which might be extended for different scales and situations, as well larger database. The article is very understandable and easy to follow. I highly recommend the acceptance of this manuscript, after minor corrections, as described below:

SPECIFIC COMMENTS

L.40-42: this statement requires a citation. Please provide it.

L.49-54: here I observed a highlighted defence for the use of machine learning algorithms. However, I suggest including the main premise to use these techniques, which is the need of a big dataset. With a small dataset, the statistical methods are more suitable (even for big data too) to solve these problems. I’m afraid that this kind of statement may mislead the audience to believe that machine learning is more preferred than statistical procedures.

L.57: just a contribution to MDPI formatting style: Change “Lu et al. (2021) …” to “Lu et al. [9] …”. Please check this problem throughout the text.

L82-88: excellent way to state a hypothesis! Congrats!

  1. 98: 15.5 km² instead of 15.5 km2.

L.108-118: please provide the sample size of this dataset. Also, how did you average the data (hourly, 2-2 hours...)?

- More information regarding the software and the environment used in this study is necessary.

- I would be pleased if the authors would make the data and algorithms (analysis script) available to the scientific community.

Author Response

(The authors gave the same response as above.)

Round 2

Reviewer 1 Report

The authors addressed all of my concerns. I have no further comment.

Reviewer 2 Report

The authors have incorporated the suggestions of the reviewer, therefore, the paper can now be accepted for publication.